# Few-Shot Learning Through an Information Retrieval Lens

**Eleni Triantafillou**
University of Toronto
Vector Institute

**Richard Zemel**
University of Toronto
Vector Institute

**Raquel Urtasun**
University of Toronto
Vector Institute
Uber ATG

## Abstract

Few-shot learning refers to understanding new concepts from only a few examples. We propose an information retrieval-inspired approach for this problem that is motivated by the increased importance of maximally leveraging all the available information in this low-data regime. We define a training objective that aims to extract as much information as possible from each training batch by effectively optimizing over all relative orderings of the batch points simultaneously. In particular, we view each batch point as a 'query' that ranks the remaining ones based on its predicted relevance to them and we define a model within the framework of structured prediction to optimize mean Average Precision over these rankings. Our method achieves impressive results on the standard few-shot classification benchmarks while is also capable of few-shot retrieval.

## 1   Introduction

Recently, the problem of learning new concepts from only a few labelled examples, referred to as few-shot learning, has received considerable attention [1, 2]. More concretely, K-shot N-way classification is the task of classifying a data point into one of N classes, when only K examples of each class are available to inform this decision. This is a challenging setting that necessitates different approaches from the ones commonly employed when the labelled data of each new concept is abundant. Indeed, many recent success stories of machine learning methods rely on large datasets and suffer from overfitting in the face of insufficient data. It is however not realistic nor preferred to always expect many examples for learning a new class or concept, rendering few-shot learning an important problem to address.

We propose a model for this problem that aims to extract as much information as possible from each training batch, a capability that is of increased importance when the available data for learning each class is scarce. Towards this goal, we formulate few-shot learning in information retrieval terms: each point acts as a 'query' that ranks the remaining ones based on its predicted relevance to them. We are then faced with the choice of a ranking loss function and a computational framework for optimization. We choose to work within the framework of structured prediction and we optimize mean Average Precision (mAP) using a standard Structural SVM (SSVM) [3], as well as a Direct Loss Minimization (DLM) [4] approach. We argue that the objective of mAP is especially suited for the low-data regime of interest since it allows us to fully exploit each batch by simultaneously optimizing over all relative orderings of the batch points. Figure 1 provides an illustration of this training objective.

Our contribution is therefore to adopt an information retrieval perspective on the problem of few-shot learning; we posit that a model is prepared for the sparse-labels setting by being trained in a manner

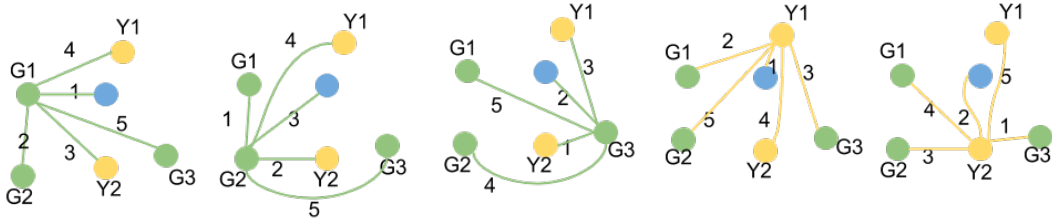

Figure 1: Best viewed in color. Illustration of our training objective. Assume a batch of 6 points: G1, G2 and G3 of class "green", Y1 and Y2 of "yellow", and another point. We show in columns 1-5 the predicted rankings for queries G1, G2, G3, Y1 and Y2, respectively. Our learning objective is to move the 6 points in positions that simultaneously maximize the Average Precision (AP) of the 5 rankings. For example, the AP of G1's ranking would be optimal if G2 and G3 had received the two highest ranks, and so on.

that fully exploits the information in each batch. We also introduce a new form of a few-shot learning task, 'few-shot retrieval', where given a 'query' image and a pool of candidates all coming from previously-unseen classes, the task is to 'retrieve' all relevant (identically labelled) candidates for the query. We achieve competitive with the state-of-the-art results on the standard few-shot classification benchmarks and show superiority over a strong baseline in the proposed few-shot retrieval problem.

## 2  Related Work

Our approach to few-shot learning heavily relies on learning an informative similarity metric, a goal that has been extensively studied in the area of metric learning. This can be thought of as learning a mapping of objects into a space where their relative positions are indicative of their similarity relationships. We refer the reader to a survey of metric learning [5] and merely touch upon a few representative methods here.

Neighborhood Component Analysis (NCA) [6] learns a metric aiming at high performance in nearest neirhbour classification. Large Margin Nearest Neighbor (LMNN) [7] refers to another approach for nearest neighbor classification which constructs triplets and employs a contrastive loss to move the 'anchor' of each triplet closer to the similarly-labelled point and farther from the dissimilar one by at least a predefined margin.

More recently, various methods have emerged that harness the power of neural networks for metric learning. These methods vary in terms of loss functions but have in common a mechanism for the parallel and identically-parameterized embedding of the points that will inform the loss function. Siamese and triplet networks are commonly-used variants of this family that operate on pairs and triplets, respectively. Example applications include signature verification [8] and face verification [9, 10]. NCA and LMNN have also been extended to their deep variants [11] and [12], respectively. These methods often employ hard-negative mining strategies for selecting informative constraints for training [10, 13]. A drawback of siamese and triplet networks is that they are local, in the sense that their loss function concerns pairs or triplets of training examples, guiding the learning process to optimize the desired relative positions of only two or three examples at a time. The myopia of these local methods introduces drawbacks that are reflected in their embedding spaces. [14] propose a method to address this by using higher-order information.

We also learn a similarity metric in this work, but our approach is specifically tailored for few-shot learning. Other metric learning approaches for few-shot learning include [15, 1, 16, 17]. [15] employs a deep convolutional neural network that is trained to correctly predict pairwise similarities. Attentive Recurrent Comparators [16] also perform pairwise comparisons but form the representation of the pair through a sequence of glimpses at the two points that comprise it via a recurrent neural network. We note that these pairwise approaches do not offer a natural mechanism to solve K-shot N-way tasks for K > 1 and focus on one-shot learning, whereas our method tackles the more general few-shot learning problem. Matching Networks [1] aim to 'match' the training setup to the evaluation trials of K-shot N-way classification: they divide each sampled training 'episode' into disjoint support and query sets and backpropagate the classification error of each query point conditioned on the support set. Prototypical Networks [17] also perform episodic training, and use the simple yet effective mechanism of representing each class by the mean of its examples in the support set, constructing a

'prototype' in this way that each query example will be compared with. Our approach can be thought of as constructing all such query/support sets within each batch in order to fully exploit it.

Another family of methods for few-shot learning is based on meta-learning. Some representative work in this category includes [2, 18]. These approaches present models that learn how to use the support set in order to update the parameters of a learner model in such a way that it can generalize to the query set. Meta-Learner LSTM [2] learns an initialization for learners that can solve new tasks, whereas Model-Agnostic Meta-Learner (MAML) [18] learns an update step that a learner can take to be successfully adapted to a new task. Finally, [19] presents a method that uses an external memory module that can be integrated into models for remembering rarely occurring events in a life-long learning setting. They also demonstrate competitive results on few-shot classification.

## 3 Background

### 3.1 Mean Average Precision (mAP)

Consider a batch $\mathcal{B}$ of points: $\mathcal{X} = \{x_1, x_2, \ldots, x_N\}$ and denote by $c_j$ the class label of the point $x_j$. Let $Rel^{x_1} = \{x_j \in \mathcal{B} : c_1 == c_j\}$ be the set of points that are relevant to $x_1$, determined in a binary fashion according to class membership. Let $O^{x_1}$ denote the ranking based on the predicted similarity between $x_1$ and the remaining points in $\mathcal{B}$ so that $O^{x_1}[j]$ stores $x_1$'s $j_{th}$ most similar point. Precision at j in the ranking $O^{x_1}$, denoted by $Prec@j^{x_1}$ is the proportion of points that are relevant to $x_1$ within the j highest-ranked ones. The Average Precision (AP) of this ranking is then computed by averaging the precisions at j over all positions j in $O^{x_1}$ that store relevant points.

$$AP^{x_1} = \sum_{\substack{j \in \{1, \ldots, |\mathcal{B}-1|: \\ O^{x_1}[j] \in Rel^{x_1}\}}} \frac{Prec@j^{x_1}}{|Rel^{x_1}|} \quad \text{where} \quad Prec@j^{x_1} = \frac{|\{k \leq j : O^{x_1}[k] \in Rel^{x_1}\}|}{j}$$

Finally, mean Average Precision (mAP) calculates the mean AP across batch points.

$$mAP = \frac{1}{|\mathcal{B}|} \sum_{i \in \{1, \ldots \mathcal{B}\}} AP^{x_i}$$

### 3.2 Structural Support Vector Machine (SSVM)

Structured prediction refers to a family of tasks with inter-dependent structured output variables such as trees, graphs, and sequences, to name just a few [3]. Our proposed learning objective that involves producing a ranking over a set of candidates also falls into this category so we adopt structured prediction as our computational framework. SSVM [3] is an efficient method for these tasks with the advantage of being tunable to custom task loss functions. More concretely, let $\mathcal{X}$ and $\mathcal{Y}$ denote the spaces of inputs and structured outputs, respectively. Assume a scoring function $F(x, y; w)$ depending on some weights w, and a task loss $L(\mathbf{y_{GT}}, \hat{\mathbf{y}})$ incurred when predicting $\hat{\mathbf{y}}$ when the groundtruth is $\mathbf{y_{GT}}$. The margin-rescaled SSVM optimizes an upper bound of the task loss formulated as:
$$\min_w \mathbb{E}[\max_{\hat{\mathbf{y}} \in \mathcal{Y}} \{L(\mathbf{y_{GT}}, \hat{\mathbf{y}}) - F(x, \mathbf{y_{GT}}; w) + F(x, \hat{\mathbf{y}}; w)\}]$$

The loss gradient can then be computed as:
$$\nabla_w L(\mathbf{y}) = \nabla_w F(\mathcal{X}, \mathbf{y}_{hinge}, w) - \nabla_w F(\mathcal{X}, \mathbf{y_{GT}}, w)$$
$$\text{with} \ \ \mathbf{y}_{hinge} = \arg\max_{\hat{\mathbf{y}} \in \mathcal{Y}} \{F(\mathcal{X}, \hat{\mathbf{y}}, w) + L(\mathbf{y_{GT}}, \hat{\mathbf{y}})\} \tag{1}$$

### 3.3 Direct Loss Minimization (DLM)

[4] proposed a method that directly optimizes the task loss of interest instead of an upper bound of it. In particular, they provide a perceptron-like weight update rule that they prove corresponds to the gradient of the task loss. [20] present a theorem that equips us with the corresponding weight update rule for the task loss in the case of nonlinear models, where the scoring function is parameterized by a neural network. Since we make use of their theorem, we include it below for completeness.

Let $\mathcal{D} = \{(x, y)\}$ be a dataset composed of input $x \in \mathcal{X}$ and output $y \in \mathcal{Y}$ pairs. Let $F(\mathcal{X}, y, w)$ be a scoring function which depends on the input, the output and some parameters $w \in \mathbb{R}^A$.

**Theorem 1** (General Loss Gradient Theorem from [20]). *When given a finite set $\mathcal{Y}$, a scoring function $F(\mathcal{X}, \mathbf{y}, w)$, a data distribution, as well as a task-loss $L(\mathbf{y}, \hat{\mathbf{y}})$, then, under some mild regularity conditions, the direct loss gradient has the following form:*

$$\nabla_w L(\mathbf{y}, \mathbf{y}_w) = \pm \lim_{\epsilon \to 0} \frac{1}{\epsilon} (\nabla_w F(\mathcal{X}, \mathbf{y}_{direct}, w) - \nabla_w F(\mathcal{X}, \mathbf{y}_w, w)) \tag{2}$$

*with:*

$$\mathbf{y}_w = \arg\max_{\hat{\mathbf{y}} \in \mathcal{Y}} F(\mathcal{X}, \hat{\mathbf{y}}, w) \quad and \quad \mathbf{y}_{direct} = \arg\max_{\hat{\mathbf{y}} \in \mathcal{Y}} \{ F(\mathcal{X}, \hat{\mathbf{y}}, w) \pm \epsilon L(\mathbf{y}, \hat{\mathbf{y}}) \}$$

This theorem presents us with two options for the gradient update, henceforth the positive and negative update, obtained by choosing the $+$ or $-$ of the $\pm$ respectively. [4] and [20] provide an intuitive view for each one. In the case of the positive update, $y_{direct}$ can be thought of as the 'worst' solution since it corresponds to the output value that achieves high score while producing high task loss. In this case, the positive update encourages the model to move away from the bad solution $y_{direct}$. On the other hand, when performing the negative update, $y_{direct}$ represents the 'best' solution: one that does well both in terms of the scoring function and the task loss. The model is hence encouraged in this case to adjust its weights towards the direction of the gradient of this best solution's score.

In a nutshell, this theorem provides us with the weight update rule for the optimization of a custom task loss, provided that we define a scoring function and procedures for performing standard and loss-augmented inference.

### 3.4 Relationship between DLM and SSVM

As also noted in [4], the positive update of direct loss minimization strongly resembles that of the margin-rescaled structural SVM [3] which also yields a loss-informed weight update rule. This gradient computation differs from that of the direct loss minimization approach only in that, while SSVM considers the score of the ground-truth $F(\mathcal{X}, \mathbf{y_{GT}}, w)$, direct loss minimization considers the score of the current prediction $F(\mathcal{X}, \mathbf{y}_w, w)$. The computation of $y_{hinge}$ strongly resembles that of $y_{direct}$ in the positive update. Indeed SSVM's training procedure also encourages the model to move away from weights that produce the 'worst' solution $y_{hinge}$.

### 3.5 Optimizing for Average Precision (AP)

In the following section we adapt and extend a method for optimizing AP [20].

Given a query point, the task is to rank N points $x = (x_1, \ldots, x_N)$ with respect to their relevance to the query, where a point is relevant if it belongs to the same class as the query and irrelevant otherwise. Let $\mathcal{P}$ and $\mathcal{N}$ be the sets of 'positive' (i.e. relevant) and 'negative' (i.e. irrelevant) points respectively. The output ranking is represented as $y_{ij}$ pairs where $\forall i, j, y_{ij} = 1$ if i is ranked higher than j and $y_{ij} = -1$ otherwise, and $\forall i, y_{ii} = 0$. Define $y = (\ldots, y_{ij}, \ldots)$ to be the collection of all such pairwise rankings.

The scoring function that [20] used is borrowed from [21] and [22]:

$$F(x, y, w) = \frac{1}{|\mathcal{P}||\mathcal{N}|} \sum_{i \in \mathcal{P}, j \in \mathcal{N}} y_{ij}(\varphi(x_i, w) - \varphi(x_j, w))$$

where $\varphi(x_i, w)$ can be interpreted as the learned similarity between $x_i$ and the query.

[20] devise a dynamic programming algorithm to perform loss-augmented inference in this setting which we make use of but we omit for brevity.

## 4 Few-Shot Learning by Optimizing mAP

In this section, we present our approach for few-shot learning that optimizes mAP. We extend the work of [20] that optimizes for AP in order to account for all possible choices of query among the batch points. This is not a straightforward extension as it requires ensuring that optimizing the AP of one query's ranking does not harm the AP of another query's ranking.

In what follows we define a mathematical framework for this problem and we show that we can treat each query independently without sacrificing correctness, therefore allowing to efficiently in parallel

learn to optimize all relative orderings within each batch. We then demonstrate how we can use the frameworks of SSVM and DLM for optimization of mAP, producing two variants of our method henceforth referred to as mAP-SSVM and mAP-DLM, respectively.

**Setup:** Let $\mathcal{B}$ be a batch of points: $\mathcal{B} = \{x_1, x_2, \ldots, x_N\}$ belonging to $\mathcal{C}$ different classes. Each class $c \in \{1, 2, \ldots, \mathcal{C}\}$ defines the positive set $\mathcal{P}^c$ containing the points that belong to $c$ and the negative set $\mathcal{N}^c$ containing the rest of the points. We denote by $c_i$ the class label of the $i_{th}$ point.

We represent the output rankings as a collection of $y^i_{kj}$ variables where $y^i_{kj} = 1$ if $k$ is ranked higher than $j$ in $i$'s ranking, $y^i_{kk} = 0$ and $y^i_{kj} = -1$ if $j$ is ranked higher than $k$ in $i$'s ranking. For convenience we combine these comparisons for each query i in $y^i = (\ldots, y^i_{kj}, \ldots)$.

Let $f(x, w)$ be the embedding function, parameterized by a neural network and $\varphi(x_1, x_2, w)$ the cosine similarity of points $x_1$ and $x_2$ in the embedding space given by $w$:

$$\varphi(x_1, x_2, w) = \frac{f(x_1, w) \cdot f(x_2, w)}{|f(x_1, w)||f(x_2, w)|}$$

$\varphi(x_i, x_j, w)$ is typically referred in the literature as the score of a siamese network.

We consider for each query $i$, the function $F^i(\mathcal{X}, y^i, w)$:

$$F^i(\mathcal{X}, y^i, w) = \frac{1}{|\mathcal{P}^{c_i}||\mathcal{N}^{c_i}|} \sum_{k \in \mathcal{P}^{c_i} \setminus i} \sum_{j \in \mathcal{N}^{c_i}} y^i_{kj}(\varphi(x_i, x_k, w) - \varphi(x_i, x_j, w))$$

We then compose the scoring function by summing over all queries: $F(\mathcal{X}, y, w) = \sum_{i \in \mathcal{B}} F^i(\mathcal{X}, y^i, w)$

Further, for each query $i \in \mathcal{B}$, we let $p^i = rank(y^i) \in \{0, 1\}^{|\mathcal{P}^{c_i}| + |\mathcal{N}^{c_i}|}$ be a vector obtained by sorting the $y^i_{kj}$'s $\forall k \in \mathcal{P}^{c_i} \setminus i, j \in \mathcal{N}^{c_i}$, such that for a point $g \neq i$, $p^i_g = 1$ if g is relevant for query i and $p^i_g = -1$ otherwise. Then the AP loss for the ranking induced by some query $i$ is defined as:

$$L^i_{AP}(p^i, \hat{p}^i) = 1 - \frac{1}{|\mathcal{P}^{c_i}|} \sum_{j : \hat{p}^i_j = 1} Prec@j$$

where $Prec@j$ is the percentage of relevant points among the top-ranked j and $p^i$ and $\hat{p}^i$ denote the ground-truth and predicted binary relevance vectors for query i, respectively. We define the mAP loss to be the average AP loss over all query points.

**Inference:** We proof-sketch in the supplementary material that inference can be performed efficiently in parallel as we can decompose the problem of optimizing the orderings induced by the different queries to optimizing each ordering separately. Specifically, for a query $i$ of class $c$ the computation of the $y^i_{kj}$'s, $\forall k \in \mathcal{P}^c \setminus i, j \in \mathcal{N}^c$ can happen independently of the computation of the $y^{i'}_{k'j'}$'s for some other query $i' \neq i$. We are thus able to optimize the ordering induced by each query point independently of those induced by the other queries. For query $i$, positive point $k$ and negative point $j$, the solution of standard inference is $y^i_{w_{kj}} = \arg\max_{y^i} F^i(\mathcal{X}, y^i, w)$ and can be computed as follows

$$y^i_{w_{kj}} = \begin{cases} 1, \text{ if } \varphi(x_i, x_k, w) - \varphi(x_i, x_j, w) > 0 \\ -1, \text{ otherwise} \end{cases} \tag{3}$$

Loss-augmented inference for query $i$ is defined as

$$y^i_{direct} = \arg\max_{\hat{y}^i} \left\{ F^i(\mathcal{X}, \hat{y}^i, w) \pm \epsilon L^i(y^i, \hat{y}^i) \right\} \tag{4}$$

and can be performed via a run of the dynamic programming algorithm of [20]. We can then combine the results of all the independent inferences to compute the overall scoring function

$$F(\mathcal{X}, y_w, w) = \sum_{i \in \mathcal{B}} F^i(\mathcal{X}, y^i_w, w) \quad \text{and} \quad F(\mathcal{X}, y_{direct}, w) = \sum_{i \in \mathcal{B}} F^i(\mathcal{X}, y^i_{direct}, w) \tag{5}$$

Finally, we define the ground-truth output value $y_{GT}$. For any query $i$ and distinct points $m, n \neq i$ we set $y^i_{GT_{mn}} = 1$ if $m \in \mathcal{P}^{c_i}$ and $n \in \mathcal{N}^{c_i}$, $y^i_{GT_{mn}} = -1$ if $n \in \mathcal{P}^{c_i}$ and $m \in \mathcal{N}^{c_i}$ and $y^i_{GT_{mn}} = 0$ otherwise.

---

**Algorithm 1** Few-Shot Learning by Optimizing mAP

---

**Input:** A batch of points $\mathcal{X} = \{x_1, \ldots, x_N\}$ of $\mathcal{C}$ different classes and $\forall c \in \{1, \ldots, \mathcal{C}\}$ the sets $\mathcal{P}^c$ and $\mathcal{N}^c$.

Initialize w
**if** using mAP-SSVM **then**
    Set $y^i_{GT} = \text{ONES}(|\mathcal{P}^{c_i}|, |\mathcal{N}^{c_i}|), \forall i = 1, \ldots, N$
**end if**
**repeat**
    **if** using mAP-DLM **then**
        Standard inference: Compute $y^i_w, \forall i = 1, \ldots, N$ as in **Equation** 3
    **end if**
    Loss-augmented inference: Compute $y^i_{direct}, \forall i = 1, \ldots, N$ via the DP algorithm of [20] as in **Equation** 4.
    In the case of mAP-SSVM, always use the positive update option and set $\epsilon = 1$

    Compute $F(\mathcal{X}, y_{direct}, w)$ as in **Equation** 5
    **if** using mAP-DLM **then**
        Compute $F(\mathcal{X}, y_w, w)$ as in **Equation** 5
        Compute the gradient $\nabla_w L(y, y_w)$ as in **Equation** 2
    **else if** using mAP-SSVM **then**
        Compute $F(\mathcal{X}, y_{GT}, w)$ as in **Equation** 6
        Compute the gradient $\nabla_w L(y, y_w)$ as in **Equation** 1 (using $y_{direct}$ in the place of $y_{hinge}$)
    **end if**
    Perform the weight update rule with stepsize $\eta$: $w \leftarrow w - \eta \nabla_w L(y, y_w)$
**until** stopping criteria

---

We note that by construction of our scoring function defined above, we will only have to compute $y^i_{kj}$'s where $k$ and $i$ belong to the same class $c_i$ and $j$ is a point from another class. Because of this, we set the $y^i_{GT}$ for each query $i$ to be an appropriately-sized matrix of ones: $y^i_{GT} = ones(|\mathcal{P}^{c_i}|, |\mathcal{N}^{c_i}|)$.

The overall score of the ground truth is then

$$F(\mathcal{X}, y_{GT}, w) = \sum_{i \in \mathcal{B}} F^i(\mathcal{X}, y^i_{GT}, w) \tag{6}$$

**Optimizing mAP via SSVM and DLM** We have now defined all the necessary components to compute the gradient update as specified by the General Loss Gradient Theorem of [20] in equation 2 or as defined by the Structural SVM in equation 1. For clarity, Algorithm 1 describes this process, outlining the two variants of our approach for few-shot learning, namely mAP-DLM and mAP-SSVM.

## 5   Evaluation

In what follows, we describe our training setup, the few-shot learning tasks of interest, the datasets we use, and our experimental results. Through our experiments, we aim to evaluate the few-shot retrieval ability of our method and additionally to compare our model to competing approaches for few-shot classification. For this, we have updated our tables to include very recent work that is published concurrently with ours in order to provide the reader with a complete view of the state-of-the-art on few-shot learning. Finally, we also aim to investigate experimentally our model's aptness for learning from little data via its training objective that is designed to fully exploit each training batch.

**Controlling the influence of loss-augmented inference on the loss gradient** We found empirically that for the positive update of mAP-DLM and for mAP-SSVM, it is beneficial to introduce a hyperparamter $\alpha$ that controls the contribution of the loss-augmented $F(\mathcal{X}, y_{direct}, w)$ relative to that of $F(\mathcal{X}, y_w, w)$ in the case of mAP-DLM, or $F(\mathcal{X}, y_{GT}, w)$ in the case of mAP-SSVM. The updated rules that we use in practice for training mAP-DLM and mAP-SSVM, respectively, are shown below, where $\alpha$ is a hyperparamter.

$$\nabla_w L(y, y_w) = \pm \lim_{\epsilon \to 0} \frac{1}{\epsilon} (\alpha \nabla_w F(\mathcal{X}, y_{direct}, w) - \nabla_w F(\mathcal{X}, y_w, w)) \quad \text{and}$$

$$\nabla_w L(y) = \alpha \nabla_w F(\mathcal{X}, y_{direct}, w) - \nabla_w F(\mathcal{X}, y_{y_{GT}}, w)$$

We refer the reader to the supplementary material for more details concerning this hyperparameter.

|  | Classification | | | | Retrieval | |
|  | 1-shot | | 5-shot | | 1-shot | |
|  | 5-way | 20-way | 5-way | 20-way | 5-way | 20-way |
|---|---|---|---|---|---|---|
| Siamese | **98.8** | 95.5 | - | - | 98.6 | 95.7 |
| Matching Networks [1] | 98.1 | 93.8 | 98.9 | 98.5 | - | - |
| Prototypical Networks [17] | **98.8** | **96.0** | 99.7 | **98.9** | - | - |
| MAML [18] | 98.7 | 95.8 | **99.9** | **98.9** | - | - |
| ConvNet w/ Memory [19] | 98.4 | 95.0 | 99.6 | 98.6 | - | - |
| mAP-SSVM (ours) | 98.6 | 95.2 | 99.6 | 98.6 | 98.6 | 95.7 |
| mAP-DLM (ours) | **98.8** | 95.4 | 99.6 | 98.6 | **98.7** | **95.8** |

Table 1: Few-shot learning results on Omniglot (averaged over 1000 test episodes). We report accuracy for the classification and mAP for the retrieval tasks.

**Few-shot Classification and Retrieval Tasks** Each K-shot N-way classification 'episode' is constructed as follows: N evaluation classes and 20 images from each one are selected uniformly at random from the test set. For each class, K out of the 20 images are randomly chosen to act as the 'representatives' of that class. The remaining $20 - K$ images of each class are then to be classified among the N classes. This poses a total of $(20 - K)N$ classification problems. Following the standard procedure, we repeat this process 1000 times when testing on Omniglot and 600 times for mini-ImageNet in order to compute the results reported in tables 1 and 2.

We also designed a similar one-shot N-way retrieval task, where to form each episode we select N classes at random and 10 images per class, yielding a pool of 10N images. Each of these 10N images acts as a query and ranks all remaining (10N - 1) images. The goal is to retrieve all 9 relevant images before any of the (10N - 10) irrelevant ones. We measure the performance on this task using mAP. Note that since this is a new task, there are no publicly available results for the competing few-shot learning methods.

**Our Algorithm for K-shot N-way classification** Our model classifies image $x$ into class $c = \arg\max_i AP^i(x)$, where $AP^i(x)$ denotes the average precision of the ordering that image $x$ assigns to the pool of all $KN$ representatives assuming that the ground truth class for image $x$ is i. This means that when computing $AP^i(x)$, the K representatives of class i will have a binary relevance of 1 while the $K(N-1)$ representatives of the other classes will have a binary relevance of 0. Note that in the one-shot learning case where K = 1 this amounts to classifying $x$ into the class whose (single) representative is most similar to $x$ according to the model's learned similarity metric.

We note that the siamese model does not naturally offer a procedure for exploiting all K representatives of each class when making the classification decision for some reference. Therefore we omit few-shot learning results for siamese when K > 1 and examine this model only in the one-shot case.

**Training details** We use the same embedding architecture for all of our models for both Omniglot and mini-ImageNet. This architecture mimics that of [1] and consists of 4 identical blocks stacked upon each other. Each of these blocks consists of a 3x3 convolution with 64 filters, batch normalization [23], a ReLU activation, and 2x2 max-pooling. We resize the Omniglot images to 28x28, and the mini-ImageNet images to 3x84x84, therefore producing a 64-dimensional feature vector for each Omniglot image and a 1600-dimensional one for each mini-ImageNet image. We use ADAM [24] for training all models. We refer the reader to the supplementary for more details.

**Omniglot** The Omniglot dataset [25] is designed for testing few-shot learning methods. This dataset consists of 1623 characters from 50 different alphabets, with each character drawn by 20 different drawers. Following [1], we use 1200 characters as training classes and the remaining 423 for evaluation while we also augment the dataset with random rotations by multiples of 90 degrees. The results for this dataset are shown in Table 1. Both mAP-SSVM and mAP-DLM are trained with $\alpha = 10$, and for mAP-DLM the positive update was used. We used $|B| = 128$ and $N = 16$ for our models and the siamese. Overall, we observe that many methods perform very similarly on few-shot classification on this dataset, ours being among the top-performing ones. Further, we perform equally well or better than the siamese network in few-shot retrieval. We'd like to emphasize that the siamese network is a tough baseline to beat, as can be seen from its performance in the classification tasks where it outperforms recent few-shot learning methods.

**mini-ImageNet** mini-ImageNet refers to a subset of the ILSVRC-12 dataset [26] that was used as a benchmark for testing few-shot learning approaches in [1]. This dataset contains 60,000 84x84 color images and constitutes a significantly more challenging benchmark than Omniglot. In order to

|  | Classification | | Retrieval | |
|  | 5-way | | 5-way | 20-way |
|  | 1-shot | 5-shot | 1-shot | 1-shot |
|---|---|---|---|---|
| Baseline Nearset Neighbors* | $41.08 \pm 0.70$ % | $51.04 \pm 0.65$ % | - | - |
| Matching Networks* [1] | $43.40 \pm 0.78$ % | $51.09 \pm 0.71$ % | - | - |
| Matching Networks FCE* [1] | $43.56 \pm 0.84$ % | $55.31 \pm 0.73$ % | - | - |
| Meta-Learner LSTM* [2] | $43.44 \pm 0.77$ % | $60.60 \pm 0.71$ % | - | - |
| Prototypical Networks [17] | $49.42 \pm 0.78$ % | **68.20** $\pm 0.66$ % | - | - |
| MAML [18] | $48.70 \pm 1.84$ % | $63.11 \pm 0.92$ % | - | - |
| Siamese | $48.42 \pm 0.79$ % | - | $51.24 \pm 0.57$ % | $22.66 \pm 0.13$ % |
| mAP-SSVM (ours) | **50.32** $\pm 0.80$ % | $63.94 \pm 0.72$ % | $52.85 \pm 0.56$ % | **23.87** $\pm 0.14$ % |
| mAP-DLM (ours) | $50.28 \pm 0.80$ % | $63.70 \pm 0.70$ % | **52.96** $\pm 0.55$ % | $23.68 \pm 0.13$ % |

Table 2: Few-shot learning results on miniImageNet (averaged over 600 test episodes and reported with 95% confidence intervals). We report accuracy for the classification and mAP for the retrieval tasks. *Results reported by [2].

compare our method with the state-of-the-art on this benchmark, we adapt the splits introduced in [2] which contain a total of 100 classes out of which 64 are used for training, 16 for validation and 20 for testing. We train our models on the training set and use the validation set for monitoring performance. Table 2 reports the performance of our method and recent competing approaches on this benchmark. As for Omniglot, the results of both versions of our method are obtained with $\alpha = 10$, and with the positive update in the case of mAP-DLM. We used $|B| = 128$ and $N = 8$ for our models and the siamese. We also borrow the baseline reported in [2] for this task which corresponds to performing nearest-neighbors on top of the learned embeddings. Our method yields impressive results here, outperforming recent approaches tailored for few-shot learning either via deep-metric learning such as Matching Networks [1] or via meta-learning such as Meta-Learner LSTM [2] and MAML [18] in few-shot classification. We set the new state-of-the-art for 1-shot 5-way classification. Further, our models are superior than the strong baseline of the siamese network in the few-shot retrieval tasks.

**CUB** We also experimented on the Caltech-UCSD Birds (CUB) 200-2011 dataset [27], where we outperform the siamese network as well. More details can be found in the supplementary.

**Learning Efficiency** We examine our method's learning efficiency via comparison with a siamese network. For fair comparison of these models, we create the training batches in a way that enforces that they have the same amount of information available for each update: each training batch $\mathcal{B}$ is formed by sampling $N$ classes uniformly at random and $|\mathcal{B}|$ examples from these classes. The siamese network is then trained on all possible pairs from these sampled points. Figure 2 displays the performance of our model and the siamese on different metrics on Omniglot and mini-ImageNet. The first two rows show the performance of our two variants and the siamese in the few-shot classification (left) and few-shot retrieval (right) tasks, for various levels of difficulty as regulated by the different values of N. The first row corresponds to Omniglot and the second to mini-ImageNet. We observe that even when both methods converge to comparable accuracy or mAP values, our method learns faster, especially when the 'way' of the evaluation task is larger, making the problem harder.

In the third row in Figure 2, we examine the few-shot learning performance of our model and the all-pairs siamese that were trained with $N = 8$ but with different $|\mathcal{B}|$. We note that for a given $N$, larger batch size implies larger 'shot'. For example, for $N = 8$, $|\mathcal{B}| = 64$ results to on average 8 examples of each class in each batch (8-shot) whereas $|\mathcal{B}| = 16$ results to on average 2-shot. We observe that especially when the 'shot' is smaller, there is a clear advantage in using our method over the all-pairs siamese. Therefore it indeed appears to be the case that the fewer examples we are given per class, the more we can benefit from our structured objective that simultaneously optimizes all relative orderings. Further, mAP-DLM can reach higher performance overall with smaller batch sizes (thus smaller 'shot') than the siamese, indicating that our method's training objective is indeed efficiently exploiting the batch examples and showing promise in learning from less data.

**Discussion** It is interesting to compare experimentally methods that have pursued different paths in addressing the challenge of few-shot learning. In particular, the methods we compare against each other in our tables include deep metric learning approaches such as ours, the siamese network, Prototypical Networks and Matching Networks, as well as meta-learning methods such as Meta-Learner LSTM [2] and MAML [18]. Further, [19] has a metric-learning flavor but employs external memory as a vehicle for remembering representations of rarely-observed classes. The experimental

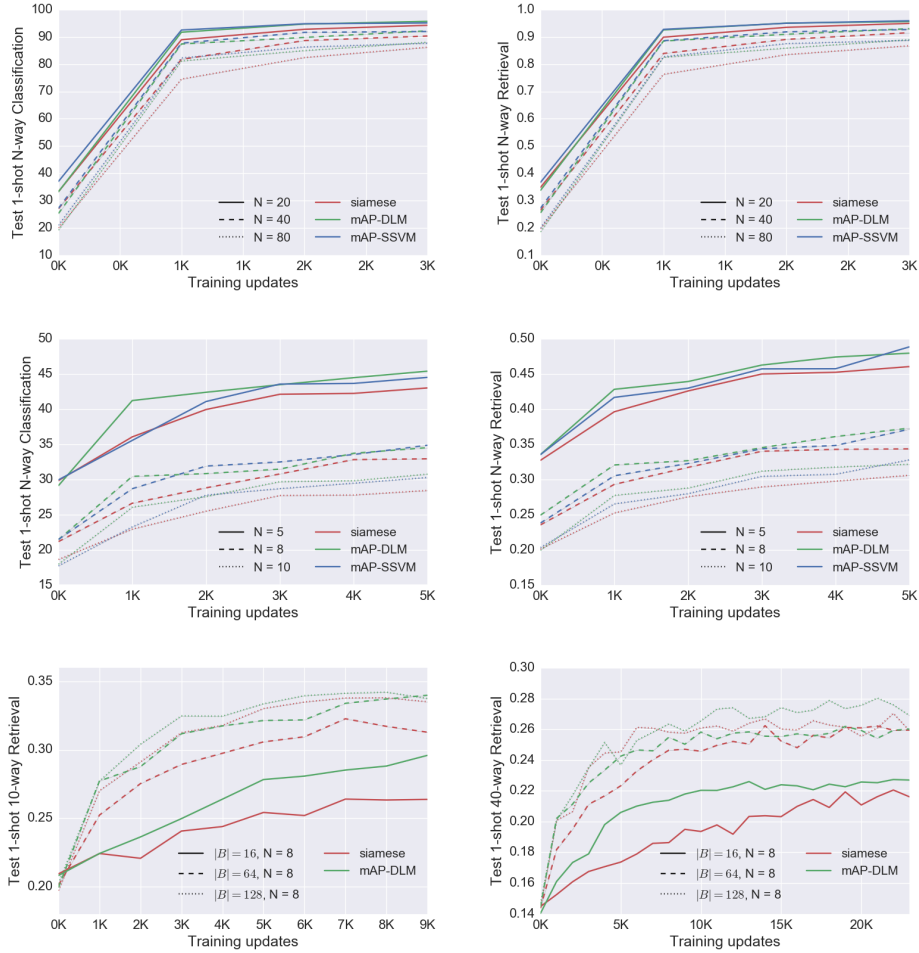

Figure 2: Few-shot learning performance (on unseen validation classes). Each point represents the average performance across 100 sampled episodes. **Top row**: Omniglot. **Second and third rows**: mini-ImageNet.

results suggest that there is no clear winner category and all these directions are worth exploring further.

Overall, our model performs on par with the state-of-the-art results on the classification benchmarks, while also offering the capability of few-shot retrieval where it exhibits superiority over a strong baseline. Regarding the comparison between mAP-DLM and mAP-SSVM, we remark that they mostly perform similarly to each other on the benchmarks considered. We have not observed in this case a significant win for directly optimizing the loss of interest, offered by mAP-DLM, as opposed to minimizing an upper bound of it.

# 6    Conclusion

We have presented an approach for few-shot learning that strives to fully exploit the available information of the training batches, a skill that is utterly important in the low-data regime of few-shot learning. We have proposed to achieve this via defining an information-retrieval based training objective that simultaneously optimizes all relative orderings of the points in each training batch. We experimentally support our claims for learning efficiency and present promising results on two standard few-shot learning datasets. An interesting future direction is to not only reason about how to best exploit the information within each batch, but additionally about how to create training batches in order to best leverage the information in the training set. Furthermore, we leave it as future work to explore alternative information retrieval metrics, instead of mAP, as training objectives for few-shot learning (e.g. ROC curve, discounted cumulative gain etc).

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
