[Supplementary Material]

# Supplementary Material

## A  Independence of Individual Query Rankings

When introducing our method, we claimed that we can decompose the problem of optimizing the mAP of the batch by optimizing the AP of each ranking independently.

We provide a proof sketch below to support this argument.

*Proof Sketch.* Let $x_i, x_k \in c_1$ be two points of class $c_1$ and $x_j \in c_2$ be a point of class $c_2$. Consider the binary rankings of the form $y_{pn}^u$ that we will have to compute involving these 3 points. We observe from the definition of the scoring function that the only $y_{pn}^u$ that will be computed are those where $u$ and $p$ are different points from the same class and $n$ is a point from a different class.

Because of this constraint, we will only have to compute $y_{kj}^i$ and $y_{ij}^k$ (we will never compute $y_{jk}^i$, $y_{ji}^k$, $y_{ik}^j$ or $y_{ki}^j$). Therefore, it remains to show that no matter what values $y_{kj}^i$ and $y_{ij}^k$ take, there is an ordering of these 3 points that satisfies them. There are four cases to consider in total since we have two variables that can each take two values.

Table 1 shows the relative positions of i, j and k in space that satisfies each of the four cases listed above, demonstrating that no contradiction can arise. □

| $y_{kj}^i$ | $y_{ij}^k$ | Configuration | | | | |
|---|---|---|---|---|---|---|
| 1 | 1 | i | k | | | j |
| 1 | -1 | i | | | k | j |
| -1 | 1 | k | | | i | j |
| -1 | -1 | i | | | j | k |

Table 1: The relative positions of points i, k, and j (configuration column) to satisfy each pairs of values of the variables $y_{kj}^i$ and $y_{ij}^k$ (specified in the first two columns).

## B  Controlling the influence of loss-augmented inference on the loss gradient

As we mentioned in the evaluation section of the paper, we found empirically that for the positive update of mAP-DLM and for mAP-SSVM, it is beneficial to introduce a hyperparamter $\alpha$ that controls the contribution of the loss-augmented $F(\mathcal{X}, y_{direct}, w)$ relative to that of $F(\mathcal{X}, y_w, w)$ in the case of mAP-DLM, or $F(\mathcal{X}, y_{GT}, w)$ in the case of SSVM.

For completeness, we review here the modified expression for the gradient computation. The updated rules that we use in practice for training mAP-DLM and mAP-SSVM, respectively, are shown below, where $\alpha$ is a hyperparamter.

$$\nabla_w L(y, y_w) = \pm \lim_{\epsilon \to 0} \frac{1}{\epsilon} (\alpha \nabla_w F(\mathcal{X}, y_{direct}, w) - \nabla_w F(\mathcal{X}, y_w, w))$$

and

$$\nabla_w L(y) = \alpha \nabla_w F(\mathcal{X}, y_{direct}, w) - \nabla_w F(\mathcal{X}, y_{y_{GT}}, w)$$

where in the case of mAP-SSVM, $\mathbf{y}_{GT}$ denotes the ground-truth labels and $\mathbf{y}_{direct} = \arg\max_{\hat{\mathbf{y}} \in \mathcal{Y}} \{F(\mathcal{X}, \hat{\mathbf{y}}, w) \pm L(\mathbf{y}, \hat{\mathbf{y}})\}$ and in the case of mAP-DLM: $\mathbf{y}_w = \arg\max_{\hat{\mathbf{y}} \in \mathcal{Y}} F(\mathcal{X}, \hat{\mathbf{y}}, w)$ and $\mathbf{y}_{direct} = \arg\max_{\hat{\mathbf{y}} \in \mathcal{Y}} \{F(\mathcal{X}, \hat{\mathbf{y}}, w) \pm \epsilon L(\mathbf{y}, \hat{\mathbf{y}})\}$.

**Exploring different values of** $\alpha$: We find experimentally that on Omniglot and mini-ImageNet, setting $\alpha > 1$ leads to significant gains, but that the value of $\alpha$ does not significantly affect performance as long as it is greater than 1. Figures 1 and 2 demonstrate this behaviour. Each point corresponds to the average few-shot learning performance over 20 randomly sampled episodes from held-out data.

a        b        c

Figure 1: Comparing the few-shot learning performance induced by different values of $\alpha$ on mini-ImageNet. **a**: 1-shot 20-way retrieval, **b**: 1-shot 5-way classification, **c**: 5-shot 5-way classification

a        b        c

Figure 2: Comparing the few-shot learning performance induced by different values of $\alpha$ on Omniglot. **a**: 1-shot 20-way retrieval, **b**: 1-shot 5-way classification, **c**: 1-shot 20-way classification

Table 2 compares the performance of mAP-SSVM and mAP-DLM for $\alpha = 1$ to their performance when $\alpha = 10$ on mini-ImageNet. There is a clear performance gap between these two settings of $\alpha$ with the latter setting producing significantly superior results. We notice that mAP-DLM suffers more than mAP-SSVM in the setting where $\alpha = 1$. We leave it to future work to explore this behavior.

| | Classification | | Retrieval | |
| | 5-way | | 5-way | 20-way |
| | 1-shot | 5-shot | 1-shot | 1-shot |
|---|---|---|---|---|
| mAP-SSVM $\alpha = 1$ (ours) | $47.89 \pm 0.78\%$ | $60.82 \pm 0.67\%$ | $52.07 \pm 0.57\%$ | $22.25 \pm 0.12\%$ |
| mAP-SSVM $\alpha = 10$ (ours) | $\mathbf{50.32} \pm 0.80\%$ | $\mathbf{63.94} \pm 0.72\%$ | $52.85 \pm 0.56\%$ | $\mathbf{23.87} \pm 0.14\%$ |
| mAP-DLM (positive) $\alpha = 1$ (ours) | $41.64 \pm 0.78\%$ | $50.40 \pm 0.64\%$ | $45.21 \pm 0.52\%$ | $17.30 \pm 0.11\%$ |
| mAP-DLM (positive) $\alpha = 10$ (ours) | $50.28 \pm 0.80\%$ | $63.70 \pm 0.70\%$ | $\mathbf{52.96} \pm 0.55\%$ | $23.68 \pm 0.13\%$ |

Table 2: Few-shot learning results on miniImageNet (averaged over 600 test episodes and reported with 95% confidence intervals). We report accuracy for the classification and mAP for the retrieval tasks.

**Relationship between** $\alpha$ **and** $\epsilon$ **(appearing in mAP-DLM)**: While these two parameters seem related, they regulate different trade-offs. $\epsilon$ determines how much to take the task loss into account when computing $y_{direct}$ while $\alpha$ regulates how much to take the score of $y_{direct}$ into account compared to the score of $y_w$ to yield the next weight update.

**Conjecture on why** $\alpha > 1$ **may help**: We conjecture that it may happen that no matter how much we choose to take the task loss into account when computing $y_{direct}$ (a choice regulated by $\epsilon$), the score of $y_{direct}$ may be similar to that of $y_w$. In this situation, because the scores of these two solutions will be very similar, by definition the task loss that we wish to minimize is close to 0 therefore providing

a weak learning signal even though the optimization is not complete. Setting $\alpha > 1$ may help in this situation to escape from this local minimum.

## C   Additional Experiments on the Caltech-UCSD Birds (CUB) 200-2011 dataset

The Caltech-UCSD Birds (CUB) 200-2011 dataset [1] contains a total of 11,788 images of 200 species of birds and is intended for fine-grained classification. This makes for a challenging few-shot learning environment since the different classes are not substantially different from each other. Following [2] we use 100 species for training, 50 for validation, and 50 for testing. All one-shot experiments are performed on species from the testing set. For each image, we take the bounding box of the bird and resize it to 3x64x64.

We construct our test procedure in the same way as described in the paper for the other datasets. For evaluation on this dataset we use the same architecture as for Omniglot and mini-ImageNet (both for the Siamese network and our model), leading to a 1024-dimensional embedding space in this case. We show our results in Table 3. We use $\alpha = 10$ in these experiments for both mAP-DLM and mAP-SSVM, and for mAP-DLM we use the positive update.

We note that the siamese network is actually a very strong baseline: it outperforms recent few-shot learning methods on mini-ImageNet and performs comparably with the state-of-the-art on Omniglot as well. Hence the 3% improvement relative to the siamese network is a significant win.

|  | Classification | | Retrieval |
|  | 1-shot 5-way | 1-shot 20-way | 1-shot 20-way |
| --- | --- | --- | --- |
| Siamese | 56.7 | 27.9 | 27.9 |
| mAP-SSVM (ours) | 59.0 | **30.8** | 30.3 |
| mAP-DLM (ours) | **59.1** | 30.6 | **30.4** |

Table 3: Few-shot classification and retrieval results on CUB (averaged over 1000 test episodes and reported with 95% confidence intervals). We report accuracy for the classification and mAP for the retrieval tasks.

## D   Training Details

Due to lack of space in the main paper, we describe the remaining training details in this section.

For Omniglot, we used a learning rate of 0.001 for mAP-DLM and mAP-SSVM, and a learning rate of 0.1 for the siamese network. For mAP-DLM, $\epsilon$ was set to 1, and for both mAP models, $\alpha = 10$. We trained all three of these models for 18K updates to achieve the optimal performance. These hyperparameter choices were based on our validation set.

For mini-ImageNet, for mAP-DLM and mAP-SSVM we used the same hyperparameters as in Omniglot, with the difference that we decay the learning rate in this case. As in Omniglot it starts from 0.001, but in this case we decay it every 2K steps starting at update 2K by multiplying it by 0.75. For the siamese network, we use a starting learning rate of 0.01 which we decay with the same schedule. These hyperparameters were set based on the performance on the validation set. We trained both mAP models for 22K update steps and the siamese for 24K update steps to achieve the best performance.