[Reviews · NeurIPS 2017]

Reviewer 1



This work is a great extension of few shot learning paradigms in deep metric learning. Rather than considering pairs of samples for metric learning (siamese networks), or 3 examples (anchor, positive, and negative -- triplet learning), the authors propose to use the full minibatchs' relationships or ranking with respect to the anchor sample. This makes sense from the structured prediction, and efficient learning. Incorporating mean average precision into the loss directly, allows for optimization of the actual task at hand. Some comments: 1. t would be great to see more experimental results, on other datasets (face recognition). 2. Figure 1 is not very powerful, it would be better to show how the distances change as training evolves. 3. Not clear how to set \lambda, as in a number of cases, a wrong value for \lambda leads to weak results. 4. Authors must provide code for this approach to really have an impact in deep metric learning, as this work requires a bit of implementation.

Reviewer 2



This paper suggests a novel training procedure for few-shot learning based on the idea of maximising information retrieval from a single batch. Retrieval is maximised by a rank-ordering objective by which one similarity ranking per sample in the batch is produced and optimised. The method and the results are clearly stated, as are the mathematical background and guarantees. On a standard Omniglot few-shot learning task the method reaches state-of-the-art results. What I am wondering about is the computational complexity of this ansatz in comparison to a Siamese or Matching Network approach. Can you report the timings for optimisation?

Reviewer 3



This paper proposes an interesting strategy for the important problem of few shot learning, comparing on relevant open benchmarks, and achieving impressive performance. However, the claims of the few shot learning being state of the art seem somewhat overstated. No reference is given to recent work that has demonstrated comparable or better results than this paper does. Relevant Papers Include: Kaiser et al., ICLR’17 Munkhdalai and Yu, ICML’17 Shyam et al., ICML’17 Overall, the approach is relatively simple from a neural network architecture perspective in comparison to these other techniques, which makes this performance of this paper impressive.